# GRANGER CAUSAL STRUCTURE RECONSTRUCTION FROM HETEROGENEOUS MULTIVARIATE TIME SERIES

## ABSTRACT

Granger causal structure reconstruction is an emerging topic that can uncover causal relationship behind multivariate time series data. In many real-world systems, it is common to encounter a large amount of multivariate time series data collected from heterogeneous individuals with sharing commonalities, however there are ongoing concerns regarding its applicability in such large scale complex scenarios, presenting both challenges and opportunities for Granger causal reconstruction. To bridge this gap, we propose a **G**ranger c**A**usal **S**tructur**E** **R**econstruction (**GASER**) framework for inductive Granger causality learning and common causal structure detection on heterogeneous multivariate time series. In particular, we address the problem through a novel attention mechanism, called prototypical Granger causal attention. Extensive experiments, as well as an online A/B test on an E-commercial advertising platform, demonstrate the superior performances of GASER.

## 1 INTRODUCTION

Broadly, machine learning tasks are either predictive or descriptive in nature, often addressed by black-box methods (Guo et al., 2018). With the power of uncovering relationship behind the data and providing explanatory analyses, causality inference has drawn increasing attention in many fields, e.g. marketing, economics, and neuroscience (Pearl, 2000; Peters et al., 2017). Since the cause generally precedes its effects, known as temporal precedence (Eichler, 2013), recently, an increasing number of studies have focused on causal discovery from time series data. They are commonly based on the concept of Granger causality (Granger, 1969; 1980) to investigate the causal relationship with quantification measures.

In many real-world systems, it is common to encounter a large amount of multivariate time series (MTS) data collected from different individuals with shared commonalities, which we define as heterogeneous multivariate time series. The underlying causal structures of such data often vary (Zhang et al., 2017; Huang et al., 2019). For example, in the financial market, the underlying causal drivers of stock prices are often heterogeneous across stocks of different plates. Similar phenomenons are also observed in the sales of different products in E-commerce. To this situation, most existing methods have to train separate models for MTS of each individual, which suffer from over-fitting especially given limited training samples. Although some works have been proposed to solve such problem (Zhang et al., 2017; Huang et al., 2019), they lack the inductive capability to do inferences for unseen samples and fall short of fully exploiting shared causal information among the heterogeneous data which often exist in practice. For instance, the causal structures of the products belonging to the same categories are usually similar. Such shared information presents opportunities for causal reconstruction to alleviate overfitting and to do inductive reasoning. However, it is also challenging to detect common and specific causal structures simultaneously.

In this paper, we propose a **G**ranger c**A**usal **S**tructur**E** **R**econstruction (**GASER**) framework for inductive Granger causality learning and common causal structure detection on heterogeneous multivariate time series data. Our approach builds on the idea of quantifying the contributions of each variable series into the prediction of target variable via a novel designed prototypical Granger causal attention mechanism. In order to ensure that the attention capturing Granger causality, we first design an attention mechanism based on Granger causal attribution of the target series and then perform prototype learning that generates both shared and specific prototypes to improve the model's robust-

ness. Extensive experiments demonstrate the superior causal structure reconstruction and prediction performances of GASER. In summary, our specific contributions are as follows:

- A novel framework that inductively reconstructs Granger causal structures and uncovers common structures among heterogeneous multivariate time series.

- A prototypical Granger causal attention mechanism that summarizes variable-wise contributions towards prediction and generates prototypes representing common causal structures.

- Relative extensive experiments on real-world, benchmark and synthetic datasets as well as an online A/B test on an E-commercial advertising platform that demonstrate the superior performance on the causal discovery and the prediction performance comparable to state-of-the-art methods.

## 2 GASER

In this section, we formally define the problem, introduce the architecture of GASER, present the prototypical Granger causal attention with the final objective function.

### 2.1 PROBLEM DEFINITION

Assuming we have a set of heterogeneous multivariate time series from $N$ individuals, i.e., $\mathbb{X} = \{\boldsymbol{X}_i\}_{i=1}^N$, with each consisting of $S$ time series of length $T$, denoted as $\boldsymbol{X}_i = (\boldsymbol{x}_i^1, \boldsymbol{x}_i^2, \ldots, \boldsymbol{x}_i^S)^\mathsf{T} \in \mathbb{R}^{S \times T}$, where $\boldsymbol{x}_i^s = (x_{i,1}^s, x_{i,2}^s, \ldots, x_{i,T}^s)^\mathsf{T} \in \mathbb{R}^T$ represents the $s$-th time series of individual $i$, and one of them is taken as the target series $\boldsymbol{y}_i$. We aim to train a model that (1) reconstructs Granger causal structures among variables for each individual; (2) generates $K$ common structures among all the $N$ individuals, each structure represented by a prototype $\boldsymbol{p}_k \in \mathbb{R}^S, k = 1, ..., K$; and (3) learns a nonlinear mapping to predict the next value of the target variable series for each individual, i.e., $\hat{y}_{i,T+1} = \mathcal{F}(\boldsymbol{X}_i)$.

### 2.2 NETWORK ARCHITECTURE

Our GASER framework consists of two parts: a set of parallel encoders, each predicting the target given the past observations, and an attention mechanism that generates prototypical Granger causal attention vectors to quantify variable-wise contributions towards prediction. Figure 1 illustrates the overall framework of GASER. As illustrated in Figure 1(a), for an input multivariate time series $\boldsymbol{X}_i$, the encoder specific to $s$-th variable projects the time series $\boldsymbol{x}_i^s$ into a sequence of hidden state, denoted as $\boldsymbol{h}_{i,t}^s = \mathcal{H}_s(x_{i,t}, \boldsymbol{h}_{i,t-1}^s)$. The encoder could be any RNN models, such as LSTM (Hochreiter & Schmidhuber, 1997) and GRU (Cho et al., 2014). The last hidden states, $\{\boldsymbol{h}_{i,T}^s\}_{s=1}^S$, are used as the hidden embeddings of each variable. Then the predicted next value of the target variable conditioned on historical data of variable $s$, denoted as $\hat{y}_{i,T+1}^s$, can be computed by $\hat{y}_{i,T+1}^s = f_s(\mathbf{h}_{i,T}^s)$, where $f_s(\cdot)$ denotes the MLP network specific to variable $s$. Then we obtain the prediction $\hat{y}_{i,T+1}$ by aggregating the predicted values specific to variables through the prototypical Granger causal attention described below.

### 2.3 PROTOTYPICAL GRANGER CAUSAL ATTENTION

We propose a novel attention mechanism in GASER, namely prototypical Granger causal attention, to reconstruct Granger causal relationships for each individual and uncover common causal structures among heterogeneous individuals. The goal is to learn attentions that can reflect the Granger causal strength between variables for each individual, and generate prototypes among heterogeneous individuals. As illustrated in Figure 1(b), the idea of the prototypical Granger causal attention mechanism is as follows. The Granger causal attribution corresponding to each individual is first computed according to the concept of Granger causality, followed by prototype learning that summarizes common causal structures for heterogeneous individuals in the training set, and produces the attention vector specific to each individual. The details of these two parts are described below.

#### 2.3.1 GRANGER CAUSAL ATTRIBUTION

Granger causality (Granger, 1969; 1980) is a concept of causality based on prediction, which declares that if a time series $\boldsymbol{x}$ Granger-causes a time series $\boldsymbol{y}$, then $\boldsymbol{y}$ can be better predicted using all available information than if the information apart from $\boldsymbol{x}$ had been used. Thus, we obtain the Granger causal attributions by comparing the prediction error when using all available information

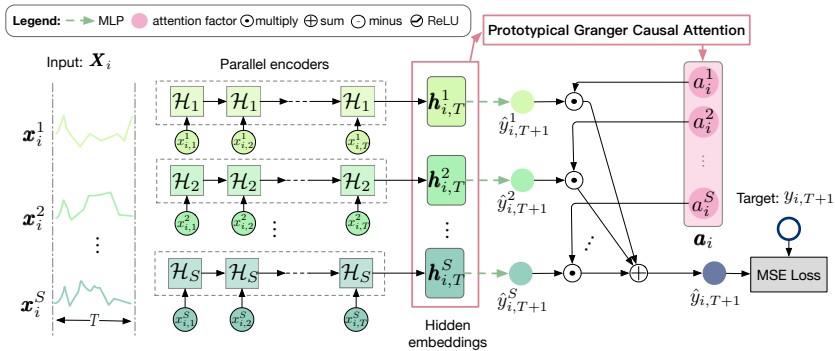

(a) The architecture of the proposed GASER.

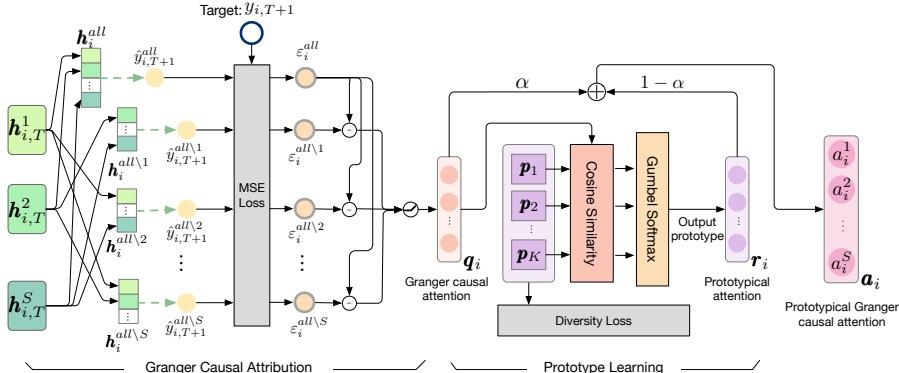

(b) Prototypical Granger causal attention.

Figure 1: The overview of the GASER framework.

with the error when using the information excluding one variable series. In particular, given all the hidden embeddings $\{\boldsymbol{h}_{i,T}^s\}_{s=1}^S$ of individual $i$, we obtain the embedding that encodes all available information and the one that encodes all available information excluding one variable $s$, denoted as $\boldsymbol{h}_i^{all}$ and $\boldsymbol{h}_i^{all\setminus s}$ respectively, by concatenating the embeddings of corresponding variables:

$$\boldsymbol{h}_i^{all} = [\boldsymbol{h}_{i,T}^j]_{j=1}^S, \quad \boldsymbol{h}_i^{all\setminus s} = [\boldsymbol{h}_{i,T}^j]_{j=1, j\neq s}^S, \tag{1}$$

where $[\cdot]$ represents the concatenation operation. Then we feed them into respective predictors, denoted as $g_{all}(\cdot)$ and $g_s(\cdot)$, to get the predicted value of target and compute the squared errors:

$$\hat{y}_{i,T+1}^{all} = g_{all}(\boldsymbol{h}_i^{all}), \quad \hat{y}_{i,T+1}^{all\setminus s} = g_s(\boldsymbol{h}_i^{all\setminus s}), \tag{2}$$

$$\varepsilon_i^{all} = (\hat{y}_{i,T+1}^{all} - y_{i,T+1})^2, \quad \varepsilon_i^{all\setminus s} = (\hat{y}_{i,T+1}^{all\setminus s} - y_{i,T+1})^2, \tag{3}$$

where the predictor $g_{all}(\cdot)$ and $g_s(\cdot)$ can be MLP networks. Inspired by Schwab et al. (2019), we define the Granger causal attribution of the target variable corresponding to variable $s$ as the decrease in error when adding $s$-th series to the set of available information, computed as:

$$\Delta\varepsilon_i^s = ReLU(\varepsilon_i^{all\setminus s} - \varepsilon_i^{all}), \tag{4}$$

where $ReLU(\cdot)$ is the rectified linear unit. For each individual $i$, by normalising the Granger causal attribution, we obtain an attention vector that reflects Granger causality, namely Granger causal attention, denoted as $\boldsymbol{q}_i$. The attention factor for variable $s$ can be computed as:

$$q_i^s = \frac{\Delta\varepsilon_i^s}{\sum_{j=1}^S \Delta\varepsilon_i^j}. \tag{5}$$

### 2.3.2 PROTOTYPE LEARNING
The Granger causal attention above is not robust enough to reconstruct Granger causal structure, given limited data (e.g., very short time series) of each individual in training. We address the problem

by generating Granger causal prototypes from all the individuals, under the assumption that there should be several common causal structures among heterogeneous individuals.

In particular, we assume there exist $K$ Granger causal prototypes, denoted as $\{\boldsymbol{p}_k\}_{k=1}^K$, and compute the similarity between the Granger causal attention vector $\boldsymbol{q}_i$ of individual $i$ and each prototype vectors $\boldsymbol{p}_k$. Since the attention can be seen as a distribution, we use the cosine similarity:

$$d_{k,i} = \frac{\boldsymbol{p}_k \cdot \boldsymbol{q}_i}{\|\boldsymbol{p}_k\| \|\boldsymbol{q}_i\|}, \tag{6}$$

Then we output a prototype most similar to $\boldsymbol{q}_i$ by sampling from the similarity distribution $\boldsymbol{d}_i$ using Gumbel-Softmax (Maddison et al., 2017; Jang et al., 2016), which samples from a reparameterized continuous distribution approximation to the categorical one-hot distribution:

$$\boldsymbol{e} = \text{GumbelSoftmax}(\boldsymbol{d}_i) = \text{softmax}((\log(\boldsymbol{d}_i) + \boldsymbol{g})/\tau), \tag{7}$$

where $\text{GumbelSoftmax}(\cdot)$ denotes the Gumbel-Softmax function, $\boldsymbol{e} \in \mathbb{R}^K$ is the sample vector which approaches one-hot, and $\boldsymbol{g}$ is a vector of i.i.d. samples drawn from $\text{Gumbel}(0,1)$ distribution. $\tau$ is the softmax temperature, and the distribution becomes discrete when $\tau$ goes to 0. With the sample vector $\boldsymbol{e}$, the output prototype $\hat{\boldsymbol{p}}$ can be obtained as:

$$\hat{\boldsymbol{p}} = [\boldsymbol{p}_1, \boldsymbol{p}_2, \ldots, \boldsymbol{p}_K] \cdot \boldsymbol{e}. \tag{8}$$

After normalizing the sampled prototype, we obtain an attention vector for individual $i$, denoted as $\boldsymbol{r}_i$, namely prototypical attention.

The Granger causal attention reflects the Granger causal structure specific to each individual, while the prototypical attention reflects one common Granger causal structure most similar to the Granger causal structure of each individual. To detect the specific and common causal structures simultaneously, we summarize them together and generate the prototypical Granger causal attention $\boldsymbol{a}_i$ as follows:

$$\boldsymbol{a}_i = \alpha \boldsymbol{q}_i + (1 - \alpha) \boldsymbol{r}_i, \tag{9}$$

where $\alpha \in [0, 1]$ is a hyperparameter that controls the ratio of the two attention mechanism.

Finally, the prediction of the target variable's next value can be computed as the weighted sum of the predicted values from all variables:

$$\hat{y}_{i,T+1} = \sum_{s=1}^S a_i^s \hat{y}_{i,T+1}^s. \tag{10}$$

## 2.4 LEARNING OBJECTIVE

In order to obtain accurate prediction and Granger causality structure, and generate diverse common causality structures, the objectives of GASER consist of three parts. The first two objective functions are to encourage accurate predictors, including the predictors $f(\cdot)$ to perform final prediction and the auxiliary predictors $g(\cdot)$ to compute Granger attribution, and we adopt the the mean squared error (MSE) as the prediction loss function:

$$\mathcal{L}_{\text{pred}} = \frac{1}{N} \sum_{i=1}^N (\hat{y}_{i,T+1} - y_{i,T+1})^2, \quad \mathcal{L}_{\text{aux}} = \frac{1}{N} \sum_{i=1}^N (\varepsilon_i^{all} + \sum_{s=1}^s \varepsilon_i^{all \backslash s}). \tag{11}$$

The last objective function is to avoid duplicate prototypes by a diversity regularization term that penalizes on prototypes that are similar to each other (Ming et al., 2019):

$$\mathcal{L}_{\text{div}} = \sum_{i=1}^K \sum_{j=i+1}^K \max(\gamma, \frac{\boldsymbol{p}_i \cdot \boldsymbol{p}_j}{\|\boldsymbol{p}_i\| \|\boldsymbol{p}_j\|}), \tag{12}$$

where $\gamma$ controls the closeness to a tolerable degree.

To summarize, the loss function, denoted by $\mathcal{L}$, is given by:

$$\mathcal{L} = \mathcal{L}_{\text{pred}} + \lambda_1 \mathcal{L}_{\text{aux}} + \lambda_2 \mathcal{L}_{\text{div}}, \tag{13}$$

where $\lambda_1$ and $\lambda_2$ are hyperparameters that adjust the ratios between the losses.

## 3 EXPERIMENTS

In this section, we evaluate the causal structure reconstruction performance on multivariate time series from both single individual and multiple individuals, as well as the prediction performance of GASER. We also conduct an online A/B test on an E-commerce advertising platform to further test GASER in more practical situations.

### 3.1 EXPERIMENTAL SETUP

We first evaluate the causal structure reconstruction performances on two causal benchmark datasets.

**Finance** (Kleinberg, 2009) consists of simulated financial market time series with known underlying causal structures. Each dataset includes 25 variables of length 4,000. For each dataset, we choose variables that are related to the most causes as the target variables to test model abilities in the relatively most challenging scenarios.

**FMRI** (Smith et al., 2011) contains 28 different Blood-oxygen-level dependent time series datasets with the ground-truth causal structures. In the experiments, we evaluate on the first 5 datasets and take the first variable as the target as causal variables distribute relatively evenly in this dataset.

Then, we evaluate the causal structure reconstruction performance on heterogeneous individuals on synthetic data:

**Synthetic data**: We first obtain the $S$ exogenous time series through the following Non-linear Autoregressive Moving Average (NARMA) (Atiya & Parlos, 2000) generators:

$$x_{i,t}^s = \alpha_s x_{i,t-1}^s + \beta_s x_{i,t-1}^s \sum_{j=1}^d x_{i,t-j}^s + \gamma_s \varepsilon_{i,t-d}\varepsilon_{i,t-1} + \varepsilon_{i,t}, \tag{14}$$

where $\varepsilon_t$ are zero-mean noise terms of 0.01 variance, $d$ is the order of non-linear interactions, and $\alpha_s$, $\beta_s$ and $\gamma_s$ are parameters specific to variable $s$, generated from $\mathcal{N}(0, 0.1)$. Then, we generate the target series from the generated exogenous series via the formula:

$$y_{i,t} = \sum_{s=1}^S \omega_i^s (\boldsymbol{\eta}_i^s)^\mathsf{T} \tanh\left(\boldsymbol{x}_{i,t-p:t-1}^s\right) + \varepsilon_{i,t}, \tag{15}$$

where $\omega_i^s \in \{0, 1\}$ with 0.6 probability of being zero that controls the underlying causal relationship from the $s$-th variable to the target variable, $\boldsymbol{\eta}_i^s \in \mathbb{R}^p$ controls the causal strength sampling from $\mathrm{Unif}\{-1, 1\}$, and $\boldsymbol{x}_{i,t-p:t-1}^s = (x_{i,t-p}^s, x_{i,t-p+1}^s, \ldots, x_{i,t-1}^s)^\mathsf{T} \in \mathbb{R}^p$ represents the last $p$ historical values of variable $s$ of sample $i$. The 0-1 indicator vector $\boldsymbol{\omega}_i = (\omega_i^1, \omega_i^2, \ldots, \omega_i^S)^\mathsf{T} \in \mathbb{R}^S$ is the ground-truth causal structure of $i$-th individual.

For the causal structure reconstruction task, we compare our method with previous causal discovery methods including linear Granger causality (Granger, 1969; Lütkepohl, 2005) and TCDF (Nauta et al., 2019), as well as the interpretable neural network based prediction method, i.e., IMV-LSTM (Guo et al., 2019), using the standard metrics of Area Under the Precision-Recall Curve (PR-AUC), and Area Under the ROC Curve (ROC-AUC) (Fawcett, 2006).

Since a byproduct of GASER is the time series prediction, we also evaluate the prediction performance on the real-world datasets, i.e., PM2.5 and SML:

**PM2.5** contains the hourly PM2.5 and meteorological data in Beijing during Jan 2010 to Dec 2014, includes 7 variables (such as PM2.5 concentration, temperature, pressure and wind speed), and forms a multivariate time series of length 43,824. The PM2.5 concentration is the target series. the dataset is split into training (60%), validation (20%) and testing sets (20%).

**SML** is a monitoring dataset for the temperature forecasting, collected from a monitor system in a domotic house for approximately 40 days. The data are sampled every minute and smoothed with the mean of every 15 minutes, forming the MTS of length 4,137. We predict the dinning-room temperature with 17 relevant variable series. The first 3,200, the following 400 and the last 537 data points are respectively used for training, validation, and test.

We compare with linear Granger causality (VAR) and the state-of-the-art prediction models including DUAL (Qin et al., 2017) and IMV-LSTM (Guo et al., 2019), and adopt Root Mean Squared Error (RMSE) and Mean Absolute Error (MAE) as metrics.

Table 1: Causal structure reconstruction results on Finance and FMRI data.

| Methods | Finance (9 datasets) | | FMRI (5 datasets) | |
|---|---|---|---|---|
| | PR-AUC | ROC-AUC | PR-AUC | ROC-AUC |
| IMV-LSTM | 0.778±0.222 | 0.862±0.172 | 0.593±0.239 | 0.620±0.136 |
| linear Granger | 0.187±0.036 | 0.652±0.084 | 0.492±0.310 | 0.654±0.126 |
| TCDF | 0.478±0.263 | 0.766±0.145 | 0.540±0.250 | 0.664±0.099 |
| GASER | **1.000±0.000**[**] | **1.000±0.000**[*] | **0.641±0.327** | **0.740±0.122** |

[**] denotes the p-value is less than 1%, and [*] denotes the p-value is less than 5%.

## 3.2 EXPERIMENTAL RESULTS

### 3.2.1 CAUSAL STRUCTURE RECONSTRUCTION PERFORMANCE ON HOMOGENEOUS MULTIVARIATE TIME SERIES

To evaluate the causal discovery performance on homogeneous multivariate time series, we train individual models for each dataset with the hyper-parameter $\alpha$ equaling 0.5. We report PR-AUC and ROC-AUC averaged across all datasets, with the standard deviation reported in Table 1. As can be seen, the proposed method greatly surpasses other methods. Especially, GASER recovers the ground-truth causal structure with high score on the Finance data.

### 3.2.2 CAUSAL STRUCTURE RECONSTRUCTION PERFORMANCE ON HETEROGENEOUS MULTIVARIATE TIME SERIES

In this part, we evaluate the causal discovery performance on heterogeneous multivariate time series. We denote the number of common causal structures as $C$, the number of variables as $S$ and the series length as $T$, and generate 100 multivariate time series for each common causal structure according to Equation (14) and Equation (15), forming $100C$ datasets. For the inductive methods GASER and IMV-LSTM, we train one model using all the datasets, while for other methods, we train separate models for each dataset. We report PR-AUC and ROC-AUC results w.r.t the variable number, the series length and the common structure number in Table 2 to 4, respectively. We observe that GASER outperforms other methods significantly in all cases, and GASER ($\alpha = 0.5$) (with the Prototypical Granger causal attention) performs better than GASER ($\alpha = 1$) (only with Granger causal attention). The observations demonstrate the superior causal discovery performance of GASER, the effectiveness of the prototypical Granger causal attention in GASER, and the advantages of utilizing shared commonalities among heterogeneous MTS. Regarding the other competitors, linear Granger performs the best followed by TCDF and IMV-LSTM at most cases. The possible reason is that linear Granger can detect Granger causal relations to some extent, though it utilizes linear model, i.e., Vector autoregression (VAR). TCDF utilizes attention-based CNN to inference potential causals followed by a causal validation step, but the attention it proposed cannot reflect Granger causality, thus achieves unsatisfactory performance. Compare to the performance on homogeneous multivariate time series, the performance of IMV-LSTM drops dramatically, which indicates that the attention mechanism in IMV-LSTM fails given heterogeneous multivariate time series.

In Table 2, we vary the number of variables to generate datasets of different complexity, and we can see that GASER outperforms other competitors consistently across different $S$, and achieves good performance when $S$ is as large as 20, demonstrating our method's capability to infer complex causal structures. Since in practice, the size of collected data is often limited, which poses challenges to recover causal structure, thus we also vary the length of time series to see the model robustness to data of small sizes. As can be seen in Table 3, GASER outperforms other methods across all cases, even when $T$ is as small as 20, which demonstrates that advantage of using shared information. We also observe that GASER ($\alpha = 0.5$) surpasses GASER ($\alpha = 1$) by a large margin, which demon-

Table 2: Causal structure reconstruction results w.r.t the variable number ($C$=3, $T$=1000).

| Methods | S=5 | | S=10 | | S=20 | |
|---|---|---|---|---|---|---|
| | PR-AUC | ROC-AUC | PR-AUC | ROC-AUC | PR-AUC | ROC-AUC |
| IMV-LSTM | 0.511±0.102 | 0.500±0.236 | 0.536±0.056 | 0.514±0.019 | 0.599±0.087 | 0.619±0.087 |
| linear Granger | 0.666±0.107 | 0.822±0.075 | 0.765±0.109 | 0.889±0.063 | 0.826±0.106 | 0.854±0.080 |
| TCDF | 0.523±0.103 | 0.523±0.214 | 0.548±0.165 | 0.587±0.180 | 0.584±0.162 | 0.642±0.152 |
| GASER ($\alpha = 1$) | 0.886±0.177[**] | 0.906±0.143[**] | 0.974±0.038[**] | 0.975±0.037[**] | 0.830±0.108 | 0.883±0.069[**] |
| GASER ($\alpha = 0.5$) | **0.911±0.147**[**] | **0.922±0.122**[**] | **0.998±0.009**[**] | **0.999±0.008**[**] | **0.858±0.103**[*] | **0.939±0.050**[**] |

[**] denotes the p-value is less than 1%, and [*] denotes the p-value is less than 5%.

Table 3: Causal structure reconstruction results w.r.t the series length ($C$=3, $S$=10).

| Methods | $T$=20 | | $T$=100 | | $T$=1000 | |
|---|---|---|---|---|---|---|
| | PR-AUC | ROC-AUC | PR-AUC | ROC-AUC | PR-AUC | ROC-AUC |
| IMV-LSTM | 0.467±0.025 | 0.541±0.035 | 0.503±0.081 | 0.511±0.018 | 0.536±0.056 | 0.514±0.019 |
| linear Granger | 0.400±0.000 | 0.500±0.000 | 0.889±0.152 | 0.943±0.085 | 0.765±0.109 | 0.889±0.063 |
| TCDF | 0.518±0.131 | 0.513±0.112 | 0.517±0.120 | 0.544±0.166 | 0.548±0.165 | 0.587±0.180 |
| GASER ($\alpha=1$) | 0.790±0.142** | 0.793±0.150** | 0.973±0.038** | 0.974±0.038** | 0.974±0.038** | 0.975±0.037** |
| GASER ($\alpha=0.5$) | **0.824±0.123**** | **0.833±0.117**** | **0.973±0.040**** | **0.976±0.036**** | **0.998±0.009**** | **0.999±0.008**** |

** denotes the p-value is less than 1%, and * denotes the p-value is less than 5%.

Table 4: Causal structure reconstruction results w.r.t. the common structure number $C$ ($S$=10, $T$=1000). We set the hyper-parameter of prototype number $K$ in the model as the same as the ground-truth common structure number $C$.

| Methods | $C$=3 | | $C$=5 | | $C$=7 | |
|---|---|---|---|---|---|---|
| | PR-AUC | ROC-AUC | PR-AUC | ROC-AUC | PR-AUC | ROC-AUC |
| GASER($\alpha=1$) | 0.974±0.038 | 0.975±0.037 | 0.891±0.118 | 0.883±0.128 | 0.838±0.118 | 0.850±0.113 |
| GASER($\alpha=0.5$) | 0.998±0.009 | 0.999±0.008 | 0.924±0.091 | 0.913±0.105 | 0.851±0.113 | 0.855±0.099 |

strates that learning prototypical attention can alleviate the over-fitting problem. In Table 4, we control the causal heterogeneity by varing the number of common causal structures $C = \{3, 5, 7\}$. We observe that the performance of GASER decreases with increasing $C$. In Figure 2, we map the learned causal attention vectors to a 2D space by the visualization tool t-SNE (Maaten & Hinton, 2008). Individuals of different causal structures are labeled by different colors. From the results, we observe that nodes belonging to the same causal structures are clustered together, which also demonstrates the effectiveness of our method.

### 3.2.3 PREDICTION PERFORMANCE

We evaluate the prediction performance on the real-world datasets, i.e., PM2.5 and SML. To evaluate the robustness of prediction and the accuracy of Granger causal attribution, we also build another datasets that only contain top 50% important variables towards prediction detected from each method. We report the prediction results in Table 5. As can be seen, GASER achieves the best performance on PM2.5 data with all features, demonstrating its superior prediction performances. We also observe that GASER achieves comparable or even better performance using selected variables, while the others' performances decrease, which indicates that effective variable selection of GASER. Linear Granger achieves the best performance on SML data, because the time series length of SML is short, thus providing limited training samples for neural-network-based methods.

### 3.3 ONLINE A/B TESTS

In order to further evaluate the effectiveness of GASER in practice, an online A/B test is conducted on an E-commercial platform, and the process is designed as follows:

- We first train GASER on the historical MTS of 30,665 items. Each MTS includes 26 variables related to searching, recommending and advertising, such as Page View (PV), Gross Merchandise Volume (GMV) and Impression Position In-Page, etc. Here, we take the item popularity as the target series, and generate the underlying causal structure for each item.

- We randomly sample 100 items whose impression position in-page Granger-causes the item popularity with high confidence, and divide them into two buckets. For Bucket A, we adjust impression positions in-page of each item by one grid since 2019/08/19 till 2019/08/29, and ensure the intervention has little impact on other variables. For Bucket B, we do nothing.

- We compare the improvement rate of item popularity week-on-week on the two bucket in Fig 3.

Table 5: Predition results under all variables and top 50% important variables on the PM2.5 and SML datasets.

| Methods | PM2.5 (all) | | PM2.5 (top 50%) | | SML (all) | | SML (top 50%) | |
|---|---|---|---|---|---|---|---|---|
| | RMSE | MAE | RMSE | MAE | RMSE | MAE | RMSE | MAE |
| linear Granger | 21.38 | 11.82 | 21.39 | 11.83 | **0.0187** | **0.0144** | **0.0190** | **0.0151** |
| DUAL | 20.93 | 11.70 | 20.90 | 11.90 | 0.0778 | 0.0676 | 0.0887 | 0.0752 |
| IMV-LSTM | 21.60 | 11.77 | 21.62 | 11.80 | 0.0747 | 0.0561 | 0.1180 | 0.0857 |
| GASER | **18.88** | **11.03** | **18.22** | **10.62** | 0.0670 | 0.0540 | 0.0621 | 0.0492 |

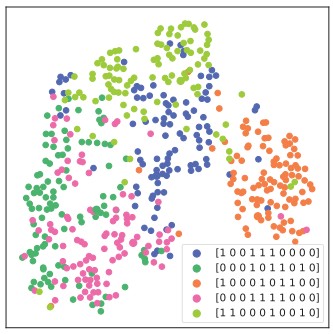 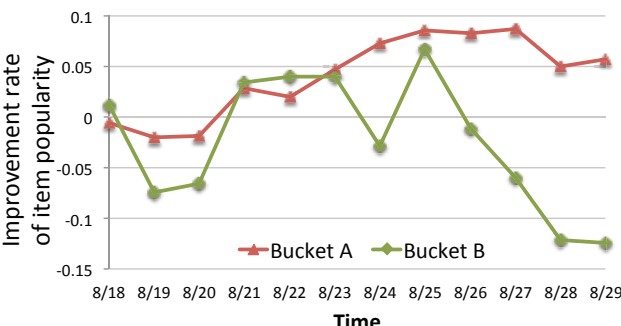

Figure 2: 2D t-SNE projections of attention vectors from 500 individuals. Color of a node indicates the underlying causal structures. The causal groundtruth $\omega_i$ is shown in the legend ($S$=10,$T$=100).

Figure 3: The result of online A/B test. The intervention starts from 08/19, and results in the item popularity improvement rate of Bucket A consistently outperforming Bucket B after 8/22, and the gap between the two buckets increases significantly since 08/25.

As shown in Figure 3, four days after the beginning of the intervention, the item popularity improvement rate of Bucket A consistently outperforms that of Bucket B, and the gap between the two buckets increases significantly since 2019/08/25, which shows that the intervention, i.e., adjusting the impression positions in-page, causes the improvement on item popularities, thus demonstrates the right causal relationships detected by GASER.

## 4 RELATED WORK

Recently a considerable amount of work has been proposed for causal inference. Classical methods, such as constraint-based methods (Pearl, 2000; Spirtes et al., 2000; Peters et al., 2013; Runge et al., 2017; Zhang et al., 2017), score-based methods (Chickering, 2002) and functional causal models (FCM) based methods (Shimizu et al., 2006), mainly focus on i.i.d data. Under the scope of time series, causal inference is commonly based on the notion of Granger causality (Granger, 1969; 1980), and a classical way is to estimate linear Granger causality under the framework of VAR models (Lütkepohl, 2005). However, existing classicial methods fail to uncover causal structures inductively. Neural network based methods that infer causal relationships or relations that approach causality have gained increasing popularity. Lopez-Paz et al. (2015) learns a probability distribution classifier to unveil causal relations. Kipf et al. (2018) proposes a neural relation inference model to infer interactions while simultaneously learning the dynamics. Yu et al. (2019) develops a deep generative model to recover the underlying DAG from complex data. Attention mechanism has often been adopted to discover relations between variables. For example, Dang et al. (2018) discovers dynamic dependencies with multi-level attention. Nauta et al. (2019) studies causal discovery through attention-based neural networks with a causal validation step. Guo et al. (2019) proposes an interpretable multi-variable LSTM with mixture attention to extract variable importance knowledge. However, these attention mechanisms provide no incentive to yield accurate attributions (Sundararajan et al., 2017; Schwab et al., 2019).

Since our method utilizes the concept of prototype to detect common causal structures, another line of related research is about prototype learning. Prototype learning is a form of cased-based reasoning (Slade, 1991), which solves problems for new inputs based on similarity to prototypical cases. Recently prototype learning has been leveraged in interpretable classification (Bien et al., 2011; Kim et al., 2014; Snell et al., 2017; Li et al., 2018; Chen et al., 2018) and sequence learning (Ming et al., 2019). We incorporate the concept for Granger causal structure reconstruction on time series data for the first time.

## 5 CONCLUSION

We formalize the problem of Granger causal structure reconstruction from heterogeneous MTS data and propose an inductive framework GASER to solve it. In particular, we propose a novel attention mechanism, namely prototypical Granger causal attention, which computes Granger causal attribution combined with prototype learning, to reconstruct Granger causal structures and uncover

common causal structures. The approach has been successfully evaluated by offline experiments on real-world and synthetic datasets compared to previous methods, also confirmed by an online A/B test on an E-commercial platform.

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

# A  TABLE OF NOTATIONS

Table 6: Terms and Notations

| Symbol | Definition |
|---|---|
| $N$ | the number of individuals |
| $S$ | the number of variables |
| $T$ | the length of time series |
| $K$ | the number of prototypes |
| $C$ | the number of underlying common causal structures |
| $\mathbb{X}$ | heterogenous multivariate time series |
| $\boldsymbol{X}_i$ | multivariate time series of individual $i$ |
| $\boldsymbol{x}_i^s$ | the $s$-th time series of individual $i$ |
| $\boldsymbol{y}_i$ | the target variable series of individual $i$ |
| $y_{i,T+1}$ | the $T+1$-step value of the target variable |
| $\hat{y}_{i,T+1}$ | the predicted $T+1$-step value of the target variable by the main predictor |
| $\hat{y}_{i,T+1}^s$ | the predicted $T+1$-step value of the target variable conditioned on variable $s$ |
| $\boldsymbol{p}_k$ | the $k$-th prototype |
| $\boldsymbol{a}_i$ | the prototypical Granger causal attention vector of individual $i$ |
| $\boldsymbol{q}_i$ | the Granger causal attention vector of individual $i$ |
| $\boldsymbol{r}_i$ | the prototypical attention vector of individual $i$ |
| $\alpha$ | the hyperparameter to control the ratio of different attentions |
| $\lambda_1, \lambda_2$ | the hyperparameter to control the ratio of loss functions |
| $\tau$ | the softmax temperature |

# B  ALGORITHM PSEUDOCODE

The full algorithm is presented in Algorithm 1. The network parameter set $\Theta$ includes the parameters of sequence encoders and MLPs. We adopt stochastic gradient descent (SGD) to optimize the network parameters and the prototype parameters. To initialize the prototypes, we first pretrain GASER for several epochs, and then employ $k$-means with cosine similarity on the Granger causal attentions $\{\boldsymbol{q}\}_{i=1}^N$, and finally we take the cluster centers as the initial prototypes. Note that the Gumbel-softmax function is only adopted in the training phase to backpropagate, and replaced by the argmax function in inference.

---

**Algorithm 1:** The algorithm of GASER.

---

**Input:**
    Input data $\mathbb{X} = \{\boldsymbol{X}_i\}_{i=1}^N$; Number of prototypes $K$; Maximum iterations $MaxIter$;
    Hyperparameters $\alpha$, $\gamma$, $\lambda_1$ and $\lambda_2$.
**Output:**
    Network parameters $\Theta$; Attention vectors $\{\boldsymbol{a}_i\}_{i=1}^N$; Prototypes $\{\boldsymbol{p}\}_{j=1}^K$; Prediction results
    $\{\hat{y}_{i,T+1}\}_{i=1}^N$.
1: Pretrain the model by optimizing $\mathcal{L}_{\text{pred}} + \lambda_1 \mathcal{L}_{\text{aux}}$;
2: Employ $k$-means on all Granger causal attention vectors $\{\boldsymbol{q}_i\}_i^N$ to get initial prototypes
    $\{\boldsymbol{p}_j\}_{j=1}^K$
3: **for** $iter \leftarrow 1$ to $MaxIter$ **do**
4:    Update network parameters $\Theta$ and prototypes $\{\boldsymbol{p}\}_{j=1}^K$ by optimizing
       $\mathcal{L} = \mathcal{L}_{\text{pred}} + \lambda_1 \mathcal{L}_{\text{aux}} + \lambda_2 \mathcal{L}_{\text{div}}$;
5: **end for**
6: Generate prototypical Granger causal attentions $\{\boldsymbol{a}_i\}_{i=1}^N$ by Equation (9) using the argmax
    function instead of the Gumbel-softmax function.

---

## C  ADDITIONAL DETAILS ON THE EXPERIMENTAL SETUP

### C.1  DATASETS

In this section, we provide some additional dataset details. **Finance** data are available at `http://www.skleinberg.org/data.html`. We use the processed **FMRI** data provide by (Nauta et al., 2019). The source and details of **PM2.5** and **SML** are at `https://archive.ics.uci.edu/ml/datasets/Beijing+PM2.5+Data` and `https://archive.ics.uci.edu/ml/datasets/SML2010`, respectively.

### C.2  IMPLEMENTATION DETAILS

We implement GASER in Tensorflow (Abadi et al., 2016) by the Adam optimizor (Kingma & Ba, 2014) with the learning rate set to 0.001. We adopt LSTMs as the sequence encoders with the hidden states size set to 128 and the window size set to 5. In all experiments, we first pretrain GASER with only the Granger causal attention for 40 epochs. The hyperparameters $\lambda_1$ and $\lambda_2$ are both set to 1, and the softmax temperature in Gumbel-softmax is set to 0.1.

### C.3  COMPARED METHODS

**Linear Granger** (Granger, 1969; 1980): We conduct a Granger causality test in the context of Vector Autoregression (VAR) as described in chapter 7.6.3 in (Lütkepohl, 2005) and implemented by the Statsmodels package (Seabold & Perktold, 2010). In detail, we perform F-test at 5% significance level. The maximum number of lags to check for order selection is set to 5, which is larger than the causal order in the ground-truth.

**TCDF** (Nauta et al., 2019): TCDF learns causal structure on multivariate time series by attention-based convolutional neural networks combined with a causal validation step. The codes are available at `https://github.com/M-Nauta/TCDF`. In all experiments, we follow the default settings as described in (Nauta et al., 2019), i.e., the significance number (stating when an increase in loss is significant enough to label a potential cause as true) as 0.8, the size of kernels as 4, dilation coefficient as 4, the learning rate as 0.01, and adopting Adam optimizor.

**DUAL** (Qin et al., 2017): It is an encoder-decoder RNN with an input attention mechanism, which forces the model pay more attention on certain driving series rather than treating all the input driving series equally. In the experiment, we use the input-attention factors to detect important variables as (Guo et al., 2019) did. We set the the size of hidden states for encoder and decoder to 64 and the window size to 10 as stated in the paper.

**IMV-LSTM** (Guo et al., 2019): It is a multi-variable attention-based LSTM capable of both prediction and variable importance interpretation, with the attention factors reflecting importance of variables in prediction. Thus, we take the learnt attention vectors as the Granger causal weights in the experiment. The codes are available at `https://github.com/KurochkinAlexey/IMV_LSTM`. In all experiments, IMV-LSTM is implemented by Adam optimizer with the mini-batch size 64, hidden layer size 128 and learning rate 0.001.

## D  SUPPLEMENTARY EXPERIMENTAL RESULTS

### D.1  VISUALIZATION OF LEARNED PROTOTYPES

We visualize the the learned prototypes and the ground-truth causal structures in Fig. 4. In this experiments, we set the hyper-parameter of prototype number $K$ equal to the ground-truth common structure number $C$. From the results, we can see that the learned prototypes are similar to the ground-truth causal structures, which demonstrates the learned prototypes are interpretable.

### D.2  VISUALIZATION OF GRANGER CAUSAL ATTENTION

In this part, we visualize the Granger causal attention vectors over epochs during the training phase and the ground-truth causal structures in Fig. 5. From the results, we can see that for the later epochs,

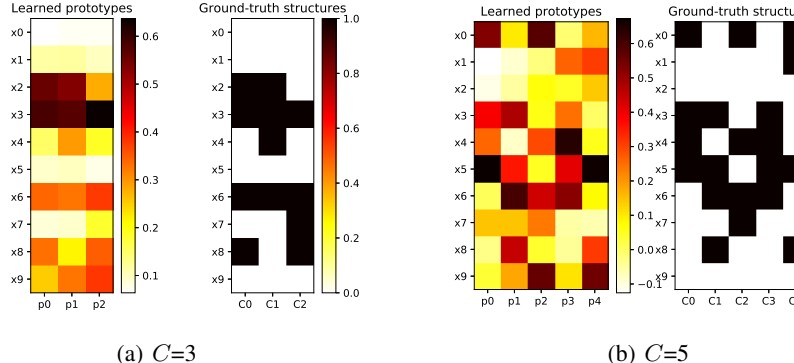

(a) $C$=3          (b) $C$=5

Figure 4: The visualization of prototypes and ground-truth causal structures. Plots are shown for various number of common causal structures $C$ ($S$=10, $T$=1000). Each column of the heat map visualizes one structure.

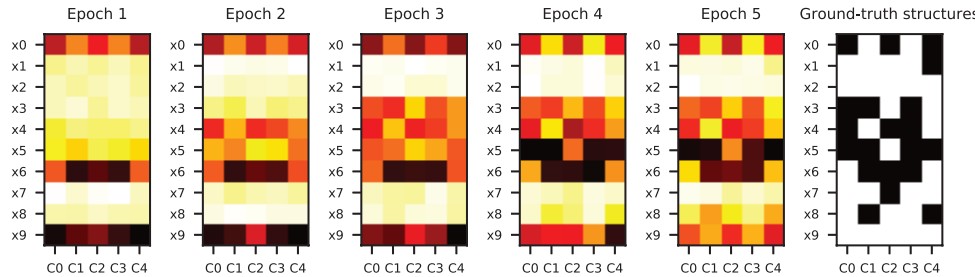

Figure 5: The visualization of Granger causal attention vectors over epochs during the training phase and the ground-truth causal structures. Each column of the heat map visualizes one structure ($C$=5, $S$=10, $T$=1000).

shown on the right of the figure, the Granger causal attention vectors are similar to the ground-truth structures. It demonstrates that the proposed Granger causal attention mechanism forces the attention values to correlate with Granger causality.

### D.3 RUNNING TIME

We compare the running time including the training time and the inference time with linear Granger method, i.e. VAR. The experiments were carried out on a server with 64 Intel(R) Xeon(R) E5-2682 v4 2.50GHz processors and 512 GB RAM. Since the linear Granger method is not inductive and has to retrain for new individuals, we set the inference time equal to the training time. As shown in Table 7, although GASER takes more time on the training phase, about 6.4 times slower than linear Granger, it is about 44.5 times faster on the inference phase.

Table 7: The comparison of running time on heterogeneous multivariate time series of 1,000 individuals ($S$=10, $T$=100, $C$=10).

|  | Training (s) | Inference (s) |
| --- | --- | --- |
| linear Granger | 19.59 | 19.59 |
| GASER | 124.53 | 0.44 |

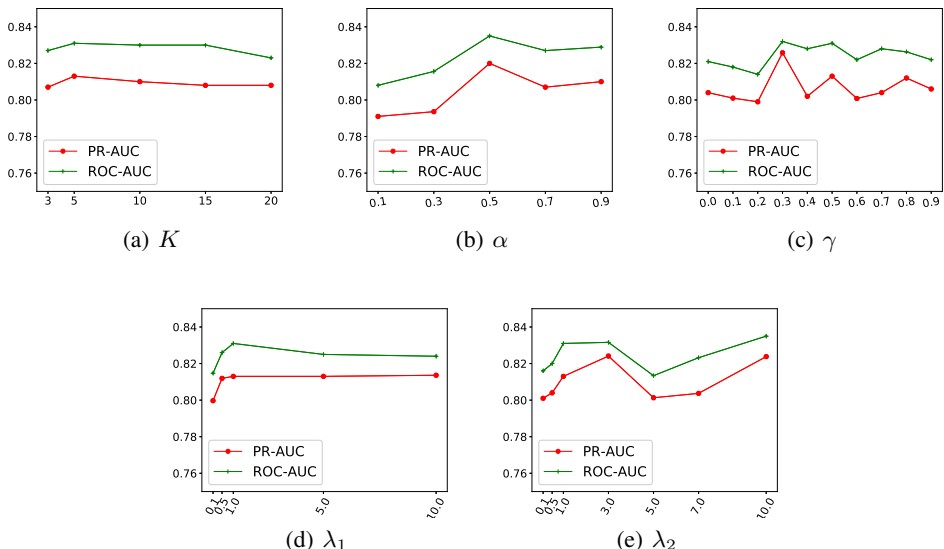

Figure 6: Performance results w.r.t. different numbers of prototypes $K$ and different values of the hyper-parameter $\alpha$, $\lambda_1$, $\lambda_2$ and $\gamma$.

### D.4 PARAMETER SENSITIVITY

We investigate the parameter sensitivity in this section. Specifically, we evaluate the sensitivity of GASER to different numbers of prototypes $K$ and different values of hyper-parameter $\alpha$,$\gamma$, $\lambda_1$ and $\lambda_2$. The results on heterogenous MTS ($C$=5, $S$=10, $T$=100) are shown in Fig. 6.

We first show how the number of prototypes affects the performance in Fig. 6(a). We can see that the performance raises when $K$ increases and achieves the best performance when the number of prototypes $K$ is equal to the number of underlying common structures $C$. Then, the performance starts to drop slowly. Overall, the proposed method is not very sensitive to the parameter $K$.

Then we evaluate how the value of $\alpha$ affects the performance in Fig. 6(b). The parameter $\alpha$ balances the weight of the Granger causal attention and the prototypical attention in our model. We can see that when $\alpha = 0.5$, the model reaches the best performance, which demonstrates both the Granger causal attention and the prototypical attention are essential in our model, and it is important to find a good balanced point between them.

The parameter $\gamma$ controls the closeness of different prototypes. The smaller the $\gamma$, the prototypes are more diverse. The result is shown in Fig. 6(c). We can see that when $\gamma$ is too small, the result is not good. This is intuitive as the ground-truth common structures among heterogeneous MTS are not totally different. Too large $\gamma$ also deteriorates the performance, because it will result in a number of similar or even duplicate prototypes. Thus, it is important to determine the parameter $\gamma$ carefully.

Figure 6(d) shows the the performance of GASER w.r.t. the parameter $\lambda_1$. The parameter $\lambda_1$ is the weight of the auxiliary prediction loss. From the result, we can see that the performance raises then $\lambda_1$ increases initially, and then keeps stable. It demonstrates that the auxiliary predictors are essential to detect the Granger causality.

Finally, we show how the value of $\lambda_2$ affects the performance in Fig. 6(e). The parameter $\lambda_1$ controls the weight of the prototype diversity regularization. From the result, we can see that the prototype diversity loss is essential in our mobel, but we should concentrate more on choosing an appropriate weight.

