# OpenReview forum: "Granger Causal Structure Reconstruction from Heterogeneous Multivariate Time Series"
_ICLR.cc/2020/Conference — Reject_

### Official Review · AnonReviewer3 · 2019-10-25
**Official Blind Review #3**

**Rating:** 6

**Review:**

The paper proposes a novel way of reconstructing Granger causal structures using a differentiable neural network architecture that contains attention modules that are proportional to the Granger causality of the input layers. Furthermore, the architecture blends individual-specific induced causal structures and cross-population prototypical causal structures. The paper has an extensive experimental section on which the proposed method shows impressive improvements in causal discovery performance and predictive performance on par with state-of-the-art.

As main contributions the paper:
* proposes a novel architecture
* shows its values using extensive experiments

Overall, I find the paper well-written with a clear description of the proposed architecture and clear experiments showing the importance of the architectural choices. A possible downside is the relative lack of novelty, since the method seems like a reasonable extension of the existing work. However, I think this counterbalanced by the excellent results on the causal discovery task and the extensive nature of the experiments.

In terms of suggestions, I think an illustrative example of the granger attention on an artificial / toy example would help a lot to give an intuitive understanding on how the method works and how the causal structure is being built.


**Experience Assessment:**

I have published one or two papers in this area.

**Review Assessment: Checking Correctness Of Derivations And Theory:**

I assessed the sensibility of the derivations and theory.

**Review Assessment: Checking Correctness Of Experiments:**

I assessed the sensibility of the experiments.

**Review Assessment: Thoroughness In Paper Reading:**

I read the paper at least twice and used my best judgement in assessing the paper.

---

> ### Author Response · Authors · 2019-11-14
> **Response to Reviewer #3**
>
> Thanks for summarizing and appreciating our work!
>
> "A possible downside is the relative lack of novelty, since the method seems like a reasonable extension of the existing work."
> > Thanks. Regarding the novelty, we think the combination of Granger causal attribution and prototype learning itself is novel and reasonable. Each module, albeit can be found in the literature, is carefully selected and re-designed to fulfill the goal of discovering Granger causality inductively from heterogeneous MTS.
>
> "In terms of suggestions, I think an illustrative example of the granger attention on an artificial / toy example would help a lot to give an intuitive understanding on how the method works and how the causal structure is being built."
> > Thanks for the suggestion! We have added a visualization of learned prototypes in Appendix D.1 and Granger attention vectors over epochs during the training phase in Appendix D.2. From the visualization, we can see that the learned prototypes and Granger attention vectors are similar to the ground-truth causal structures, which demonstrates the learned prototypes and attention vectors are interpretable.

---

### Official Review · AnonReviewer4 · 2019-11-01
**Official Blind Review #4**

**Rating:** 3

**Review:**

This paper proposes a new way of finding the Granger temporal-causal network based on attention mechanism on the predictions obtained by individual time series. It describes a surprisingly complex procedure for computing the attention vector based on combining Granger-inspired attentions with attentions obtained during a diverse prototype generation process. There are also extensive experiments demonstrating the success of the proposed method in uncovering the underlying temporal-causal graph.

There is a major theoretical and conceptual issue in this paper: the proposed attention vector depends on all of the time series. Thus, theoretically, it is a mistake to think that the $i$th time series causes the target if its attention value is higher.  In other words, the attention values leak the information about all time series in prediction of the target time series. The following paper is helpful on this topic:

Sarthak Jain and Byron C Wallace. "Attention is not explanation". In NAACL 2019.

The authors should be extremely careful in terms of calling Granger causality the actual causality. Granger causality only discovers time order. Also, it is unclear why the authors use terms such as "inductive" Granger causality, which is redundant.

It is unclear for me if the results in the paper are statistically significant, given the provided intervals.

The authors should expand the motivation for the prototypical design. In the experiments they show that having the prototype part improves robustness and accuracy, but the authors never show what interesting prototypes they learn. Also, this model is super complex. A key reason for popularity of the Granger causality is the simplicity of the underlying VAR model. Finally, the authors need to provide the run-time speed of training and inference for the model.



**Experience Assessment:**

I have published in this field for several years.

**Review Assessment: Checking Correctness Of Derivations And Theory:**

I carefully checked the derivations and theory.

**Review Assessment: Checking Correctness Of Experiments:**

I assessed the sensibility of the experiments.

**Review Assessment: Thoroughness In Paper Reading:**

I read the paper at least twice and used my best judgement in assessing the paper.

---

> ### Author Response · Authors · 2019-11-14
> **Response to Reviewer #4 (part 1)**
>
> Thanks for the valuable comments and suggestions. It seems that there are some misunderstandings due to some presentation issues and we hope the following responses address the concerns. Reviewer #4 points out that "There is a major theoretical and conceptual issue in this paper: the proposed attention vector depends on all of the time series. Thus, theoretically, it is a mistake to think that the i-th time series causes the target if its attention value is higher.  In other words, the attention values leak the information about all time series in prediction of the target time series."  We would like to clarify the misunderstandings and  point out a few key points:
>
> 1. The topic on attention and its interpretation is still a controversial and inconclusive topic. Wiegreffe S et al. challenged the paper "Attention is not explanation"([1]) and claimed that attention can be used as explanation in [8].  And Byron C Wallace made a response in https://medium.com/@byron.wallace/thoughts-on-attention-is-not-not-explanation-b7799c4c3b24. So the topic is still under heated discussion now.
>
> 2. We understand and fully agree that Granger causality [11] is not the actual causality. Therefore, we only claim that the proposed method is for Granger causality discovery throughout the paper. Granger causality is based on the following two fundamental principles: (i) the cause happens prior to its effect. (ii) the cause has information about the future values of its effect [10]. Thus, Granger causality detects important variables to the prediction of the target variable, and in this way, it can be used to find the potential causes (may be false positive) of the target variable. Also, as discussed in [10], Granger causality can be an essential tool to discover (at least partially) the causal structure if used in the right way.
>
> 3. We have concerns regarding the comment that "...the proposed attention vector depends on all of the time series...the attention values leak the information about all time series in prediction of the target time series". Yes, shared or overlapping attention mechanism  (such as [4] and [5]) may lead to information leakage across different variables; however, the proposed attention mechanism is different. The proposed Granger causal attention does not depend on all of the time series. Instead, the core idea is to train distinct predictors on distinct subsets of the input data, i.e., all available information excluding one variable $h_i^{all\backslash{s}}$, to measure how much the exclusion of individual variables reduces the prediction performance. The decrease in error is similar to the leave-one-out feature importance measure, which is claimed as an interpretable measure in [1]. And this is consistent with the definition of Granger causality in [6], thus we name it Granger causal attention. Also, a similar method has been proposed in [3], which demonstrates the capability of feature importance estimation of the attention mechanism.
>
> 4. Regarding the reference [1] the reviewer mentioned, it mainly claimed attention is not explanation as the attention weights are frequently uncorrelated with measures of feature importance. We fully agree that the traditional na¨ıve soft attention mechanisms do not necessarily force attention values to correlate with feature importance to the target as [1][2][3] claimed, thus cannot be used to uncover Granger causality. That is why we design a new attention mechanism to ensure the assigned attention weights which can assign higher value to the variable if it Granger-causes the target, and it corresponds to a sense of the feature importance measure in [1].
>
> References:
> [1] Sarthak Jain and Byron C Wallace. "Attention is not explanation". In NAACL 2019.
> [2] Sundararajan, M.; Taly, A.; and Yan, Q. "Axiomatic attribution for deep networks". In ICML 2017.
> [3] Schwab P, Miladinovic D, Karlen W. "Granger-causal attentive mixtures of experts: Learning important features with neural networks". In AAAI 2019.
> [4] Xu K, Ba J, Kiros R, et al. "Show, attend and tell: Neural image caption generation with visual attention". In ICML 2015.
> [5] Yang, Z.; Yang, D.; Dyer, C.; He, X.; Smola, A. J.; and Hovy, E. H. "Hierarchical Attention Networks for Document Classification". In NAACL 2016.
> [6] Granger, C.W. 1969. "Investigating causal relations by econometric models and cross-spectral methods." Econometrica: Journal of the Econometric Society.
> [7] Hamilton W, Ying Z, Leskovec J. "Inductive representation learning on large graphs". In NIPS 2017.
> [8] Wiegreffe S, Pinter Y. "Attention is not not Explanation". In EMNLP 2019.
> [9] Snell J, Swersky K, Zemel R. "Prototypical networks for few-shot learning" in NIPS 2017.
> [10] Eichler, Michael (2012). "Causal Inference in Time Series Analysis" In Berzuini, Carlo (ed.). Causality : statistical perspectives and applications (3rd ed.). Hoboken, N.J.: Wiley. pp. 327–352. ISBN 978-0470665565.
> [11] https://en.wikipedia.org/wiki/Granger_causality

---

> > ### Author Response · Authors · 2019-11-14
> > **Response to Reviewer #4 (part 2)**
> >
> > Other concerns:
> >
> > Q: "The authors should be extremely careful in terms of calling Granger causality the actual causality."
> > A: We understand and fully agree that Granger causality is not the actual causality. Therefore, we only claim that the proposed method is for Granger causality discovery throughout the paper.
> >
> > Q: "it is unclear why the authors use terms such as "inductive" Granger causality, which is redundant."
> > A: Sorry for the unclear presentation. The term "inductive"  here is the concept in the machine learning area. The goal of inductive learning is to learn a function that can support inference for new data samples not seen in the training set. Similar usage of "inductive" can be seen in [7]. Traditional Granger causality methods, e.g. VAR, are not inductive, i.e., they train separate and independent models for each individual. When facing a large amount of MTS from different individuals, they have to train a great many models. Moreover, each model is trained with the data from one individual, suffering from over-fitting especially given limited training samples of the individual. To address the problem, we propose the inductive Granger causal structure reconstruction model, it can generalize to unseen individual and fully exploit shared causal information among different individuals. So the term "inductive" is not redundant but reflecting our novelty.
> >
> > Q: "It is unclear for me if the results in the paper are statistically significant, given the provided intervals."
> > A: Thanks for the suggestion. Actually, GASER (α = 0.5) (with the Prototypical Granger causal attention) and GASER (α = 1) (only with Granger causal
> > attention) are both our methods. From the results, on the homogeneous MTS experiments, we can see that GASER outperforms the best results of other baselines by about 28.5% of PR-AUC and 16.0% of ROC-AUC on Finance data, and 7.9% of PR-AUC and 11.4% of ROC-AUC on FMRI data. On the heterogeneous MTS experiments, GASER  provides average gains by about 23.7% of PR-AUC and about 11.5% of ROC-AUC. We have also provided a significant level through p-value in Table 1, Table 2 and Table 3 on the updated version, which demonstrates the results are statistically significant in most cases.
> >
> > Q: "this model is super complex."
> > A: Yes, our method is more complex than traditional Granger causality methods. However, the complex parts of our model, e.g., parallel LSTMs, Granger causal attribution and prototypical learning, are necessary. This is because we aim to explore temporal nonlinearity and design an inductive model to support inference for a great many individuals in one model, and the goal can not be achieved by linear and simple models. Thus, we utilize LSTMs to capture nonlinearity, and design the prototypical Granger causal attention to inductively reconstruct Granger causal structure for multiple individuals. While simple Granger causality models such as VAR have to train separate models for different individuals, and VAR can only capture linear relationships. In this respect, we don't think our model is super complex.
> >
> > Q: "The authors should expand the motivation for the prototypical design. In the experiments they show that having the prototype part improves robustness and accuracy, but the authors never show what interesting prototypes they learn."
> > A: Thanks for the suggestion. The prototypical is to improve the stability of inductive learning,   where the model must generalize to unobserved samples, thus supporting processing a massive number of test samples not seen in the training data in a short time. To achieve this goal, we utilize the prototype learning based on the idea there exist shared and common Granger causal structures among different individuals: we first summarize some prototypes given the training individuals, and then draw conclusions for new inputs by comparing them with the prototypes. A similar idea of prototype learning can be seen in [9]. We have expanded the motivation for the prototypical design in the updated version and visualized the prototypes we learn in Appendix D.1.
> >
> > Q: "the authors need to provide the run-time speed of training and inference for the model."
> > A: Thanks for the suggestions, we have provided the run-time speed of training and inference for the model in Appendix D.3 in the updated version.
> >                         | Training (s)  |   Inference (s)
> > linear Granger|     19.59      |     19.59
> > GASER              |    124.53     |      0.44
> > Since the linear Granger method (VAR) is not inductive and has to retrain for new individuals, its inference time is equal to the training time. From the running time results, we can see that although GASER takes more time than VAR in the training phase (about 6 times slower), it takes less time in the inference phase (about 45 times faster). Thus, Gaser supports processing a massive number of test samples not seen in the training data in a short time.

---

> > ### Public Comment · ~Xinke_Shen1 · 2020-01-03
> > **How do you implement testing in prediction tasks?**
> >
> > It's a very good job. The performance is very impressive. I have a question about the testing in series prediction task: y_(i, T+1) is used in calculating the attention parameters a_i. However, obviously, you don't know y_(i, T+1) in testing. Then how do you determine a_i? Did you fix the attention parameters in testing as prototypes? I'm just very curious about this detail. I'm looking forward to your reply. Thanks!

---

### Official Review · AnonReviewer2 · 2019-11-04
**Official Blind Review #2**

**Rating:** 8

**Review:**

--
Comments after reading the reply from the authors:
Thanks for the clarification which resolves most of my concerns, and I have updated the score accordingly.  Besides, it will be more convincing to add more baselines for comparison or provide more explanation why linear Granger (which is proposed decades ago) is a strong baseline.

--
This paper investigates the important problem of inferring Granger casual structures from multi-variate time-series data, and propose the Granger casual structure reconstruction (GASER) framework with prototypical Granger causal attention.  The paper is in general well written with clear notations and is easy to follow. The proposed attention mechanism is also  intuitive. Experiments on  both simulated/synthetic and real-world datasets show the proposed approach achieved both improved casual recovery and more accurate predictions.

Here are some concerns, and the reviewer is willing to adjust the rating if these concerns are resolved.

- Baselines. Have enough strong baseline algorithms been included?
 (1) It seems that GASER outperforms state-of-the-art prediction algorithm IMV-LSTM by a large margin (Table 5) even if it is not designed for the prediction task.
(2) In Table 1-3, the other two baselines usually perform similarly or worse than the simple linear Granger baseline. Does it indicate the compared baselines are not strong enough?
(3) Why linear Granger, i.e, VAR with L1 regularization, is not included in Table 5 for prediction?

- Model design and trade-off.  According to Table 1-5, it seems that by adding the proposed attention mechanism, we can achieve both improved results (better than SOTA IMV-LSTM) and better interpretability.
(1) Is there any trade-off in the design?
(2) Is the proposed approach sensitive to different parameters, e.g., 1) the number of prototypes, $K$ (when it is not the same with the ground-truth), 2) $\alpha$ in Equation 9 (is 0.5 is a generally good default choice?), 3) $\lambda_1, \lambda_2$ in Equation 13, 4) $\gamma$ in Equation 12.

- Prototype learning. It seems that the prototype learning is used to deal with the lack of data. However, adding prototype learning increases the number of parameters to be learnt, i.e., $[p_1, … p_K]$.  It will be helpful to provide more intuition.

**Experience Assessment:**

I have published one or two papers in this area.

**Review Assessment: Checking Correctness Of Derivations And Theory:**

I assessed the sensibility of the derivations and theory.

**Review Assessment: Checking Correctness Of Experiments:**

I carefully checked the experiments.

**Review Assessment: Thoroughness In Paper Reading:**

I read the paper at least twice and used my best judgement in assessing the paper.

---

> ### Author Response · Authors · 2019-11-15
> **Response to Reviewer #2**
>
> Thank you for the detailed comments and positive feedback. We hope that our responses below resolve your concerns.
>
> Q1(1):  "It seems that GASER outperforms state-of-the-art prediction algorithm IMV-LSTM by a large margin (Table 5) even if it is not designed for the prediction task."
> A: First, I think IMV-LSTM [1], published in ICML 2019, is a relatively strong baseline, because its release time is very close to now, and it compared with comprehensive baselines, including statistics baselines and machine learning baselines. We believe it is normal that GASER also has good prediction performance. Firstly, our model makes a prediction based on Granger-causality, which is closer to true causality and more robust than the correlations especially when the training distribution and the test distribution are not exactly the same. Secondly, the task of detecting Granger causality itself requires good prediction performance as the basis, so our model is not designed without considering predictive performance.
>
> Q1(2): "In Table 1-3, the other two baselines usually perform similarly or worse than the simple linear Granger baseline. Does it indicate the compared baselines are not strong enough?"
> A: Actually, we think the linear Granger is a strong baseline in Granger causality discovery, and it is commonly used in statistics. We compare GASER with the other two baselines because we all use attention-based neural networks, and they are relatively new methods (published in 2019). They show similar or worse performance because their attention mechanisms do not force attention values to capture granger causality.
>
> Q1(3): "Why linear Granger, i.e, VAR with L1 regularization, is not included in Table 5 for prediction?"
> A: Thanks for the suggestion. Actually, we thought VAR is not strong enough in prediction due to its temporal linearity, so we didn't compare it with VAR in the prediction experiments. Now, we have added the prediction performance of VAR in Table 5.
>
> Q2(1): "Is there any trade-off in the design?"
> Yes, our model can inductively reconstruct Granger causal structures and discover common structures among different individuals, but we need to set the prototype number $K$ and the diversity parameter $\gamma$ properly. And if there exist no common structures among individuals, the prototype learning is not helpful; however, this can be addressed by setting larger $\alpha$ i.e., set a smaller weight on the prototypical attention.
>
> Q2(2): "Is the proposed approach sensitive to different parameters?"
> A: Thanks for the suggestions. We have added the parameter sensitivity experiments in Appendix D.4. From the results, we found that GASER is not very sensitive to parameters.
>
> Q3: "It seems that the prototype learning is used to deal with the lack of data. However, adding prototype learning increases the number of parameters to be learnt"
> A: Thanks for the interesting comments. It is not accurate to say that the prototype learning is used to deal with the lack of data. Actually, the prototype learning is to improve the stability of inductive learning given limited data of EACH individual, by summarizing knowledge from the data of all individuals. In other words, although the data of each individual are not enough, the whole data of all the training individuals are enough to support prototype learning since we face a large number of individuals.
>
> Reference:
> [1] "Exploring Interpretable LSTM Neural Networks over Multi-Variable Data". in ICML 2019

---

### Decision · Program_Chairs · 2019-12-19

**Decision:**

Reject

**Comment:**

This paper proposes a solution to learn Granger temporal-causal network for multivariate time series by adding attention named prototypical Granger causal attention in LSTM.

The work aims to address an important problem. The proposed solution seems effective empirically. However, two major issues have not been fully addressed in the current version: (1) the connection between Granger causality and the attention mechanism is not fully justified; (2) the complex design overkills the whole concept of Granger causality (since its popularity is due to the simplicity).

The paper would be a strong publication in the future if the two issues can be addressed in a satisfactory way.